# Does Motor Imagery Training Improve Service Performance in Tennis Players? A Systematic Review and Meta-Analysis

**DOI:** 10.3390/bs14030207

**Published:** 2024-03-05

**Authors:** Nuannuan Deng, Kim Geok Soh, Borhannudin Bin Abdullah, Dandan Huang

**Affiliations:** 1Department of Sports Studies, Faculty of Educational Studies, University Putra Malaysia, Serdang 43400, Malaysia; dengnuannuan117@gmail.com (N.D.); kims@upm.edu.my (K.G.S.); borhannudin@upm.edu.my (B.B.A.); 2College of Physical Education, Chong Qing University, Chongqing 400044, China

**Keywords:** motor imagery, mental training, tennis service, sports psychology, performance

## Abstract

Motor imagery training is a common mental strategy used by tennis players and coaches to improve learning and performance; however, the effect of motor imagery training on service performance in tennis players is questionable. This review aims to consolidate existing research regarding the effects of motor imagery training on the service performance of tennis players. A systematic search was conducted following the PRISMA guidelines, using PubMed, Web of Science, SCOPUS, and SPORTDiscus to identify articles published until December 2023. Eligible studies comprised controlled trials that investigated the impact of motor imagery on service performance outcomes in tennis players. The methodological quality of individual studies was assessed using the Cochrane RoB-2 and ROBINS-I tools. GRADE was applied to assess the certainty of the evidence. Nine trials including 548 participants met the inclusion criteria. The results indicated that motor imagery training improved service accuracy and technique but did not affect service speed or return accuracy in tennis players. In conclusion, the certainty of the evidence that motor imagery training may be effective in improving service accuracy and technique in tennis players is low to very low. However, more experimental work is needed to obtain stronger conclusions.

## 1. Introduction

In tennis, every point begins with the service, making it the pivotal shot and a crucial determinant of success in the game [1]. A scientific finding has pointed out that approximately 25% of tennis scores originate from the service [2]. However, mastering the service poses a formidable challenge, given the intricate coordination of limb and joint movements needed to accumulate and transfer forces effectively from the ground up to the racquet head. This process is referred to as the kinetic chain [3]. The acquisition of proficient skills is frequently credited to extensive periods of physical training, demanding high levels of attention and effort [4,5]. An alternative that sports psychologists and coaches have widely used to improve athletic performance is motor imagery training [6,7]. Motor imagery entails mentally envisioning an action without physically carrying it out [8,9], triggering activity in brain regions typically engaged during the actual execution of the task [10]. Motor simulation theory proposes that imagined actions are functionally equivalent to actual ones in terms of intention, planning, and the involvement of neural pathways [11,12,13]. Indeed, motor imagery training can aid in the development of motor expertise, which in turn facilitates activity-dependent neural reorganizations in the brain, controlling both imagined and real performances [14]. From a practical standpoint, the application of motor imagery practice plays an important role in various sports, contributing to the acquisition of motor skills, and preparation before competitions [15,16]

The effectiveness of motor imagery training in learning and performance improvement is supported by previous meta-analyses. Behrendt et al. [17] concluded in their meta-analytic study that motor imagery practice helps improve the motor learning of children and adolescents. Lindsay et al. [7] emphasize that motor imagery training could serve as an effective strategy for cultivating sport-specific motor skills. Simonsmeier et al. [18] found that imagery interventions notably improved motor performance, motivation, and affective outcomes in sports. On the other hand, many individual studies have suggested that motor imagery intervention has positive effects on skill performance among athletes. For example, Fazel et al. [19] highlight the positive impact of employing motor imagery to improve free-throw shooting performance in basketball players. Fortes et al. [20] concluded that imagery training enhances the passing decision-making performance of volleyball players. Björkstrand & Jern [21] reported that an imagery intervention can improve the penalty-taking ability of soccer players.

Recently, there has been increasing interest in examining the impact of motor imagery practice on tennis service performance. However, the results and conclusions drawn were partly contradictory. For example, although studies indicated improved service speed following motor imagery training [22,23], researchers in other investigations did not observe an increase in service speed after motor imagery training [24,25]. Therefore, there is a need for thorough meta-analyses to consolidate findings from previous studies and investigate whether motor imagery training could yield performance benefits for tennis players, specifically in terms of service. The present meta-analysis seeks to provide a comprehensive, evidence-based synthesis of the efficacy of motor imagery training programs in enhancing tennis service performance. It addresses the question: Does motor imagery training improve service performance in tennis players compared to controls? Consequently, we hope that this meta-analysis offers novel and valuable insights into the impact of motor imagery on service outcomes in tennis.

## 2. Materials and Methods

The present review is reported following the updated PRISMA statement [26], and the review protocol has been registered in PROSPERO (identifier CRD42023490406).

### 2.1. Search Strategy and Study Selection

A comprehensive search using specific keywords was conducted on four databases (PubMed, Web of Science, SCOPUS, and SPORTDiscus) to identify peer-reviewed journal articles in English from the databases’ inception until 6 December 2023. Boolean search strategies incorporated the following search terms: (“motor imagery” OR “mental training” OR “movement imagery” OR “mental practice” OR “mental simulation” OR “cognitive training” OR “mental imagery” OR “mental rehearsal” OR “mental movements” OR “visual imagery” AND tennis. Furthermore, we manually searched Google Scholar and the reference lists of all chosen papers to ensure that no pertinent publications were overlooked. In cases where the full text of a publication was not available, we contacted the corresponding author via email or ResearchGate. Appendix A contains the search strings used for each database.

As depicted in Figure 1, the databases provided a total of 280 publications, and we obtained an additional 10 papers through references and Google Scholar. After the manual removal of duplicates, there were 206 unique records remaining. The titles and abstracts of these records were assessed, resulting in 46 publications deemed suitable for full-text examination. After a thorough review of all the texts, 37 documents were excluded, leaving nine trials that fulfilled all the criteria established for the systematic review and meta-analysis.

### 2.2. Eligibility Criteria

The inclusion criteria were chosen according to the PICOS approach [27]: (a) The study recruited female and/or male tennis players in any age category. (b) The study’s training group participated in motor imagery practice. Motor imagery could be implemented as an independent intervention or in conjunction with physical practice. Motor imagery interventions that included the combination of motor imagery and action observation (e.g., observing a video) were also included. (c) The study included a control group. (d) The study included at least one measure of tennis service performance at baseline and follow-up. (e) The study was a randomized controlled trial (RCT) or non-RCT.

The studies were excluded if (a) the tennis players who participated in the study were injured; (b) the motor imagery intervention was combined with other aerobic training in order to avoid contamination of the motor imagery effect from other interventions; (c) the study lacked a control group; and (d) the study lacked sufficient data to compute the effect size (ES).

### 2.3. Data Extraction

Two reviewers (ND and DH) were tasked with extracting participant information, including sample size, sex, age, training experience, competition level, a description of the intervention, and study outcomes. In studies presenting data in visual formats such as graphs or figures, the reviewers either reached out to the corresponding author to get the numerical data for analysis or used the Web Plot Digitizer program (DigitizeIt, Version 4.6, Germany) to extract the required data [28]. A third reviewer (KGS) was responsible for verifying all the extracted information and data.

### 2.4. Risk of Bias in Individual Studies and Certainty of the Evidence

The updated Cochrane Risk of Bias assessment for randomized trials (RoB-2) was utilized to evaluate the bias risk in each of the selected RCTs [29]. Using the RoB 2.0 tool, studies were awarded an overall risk of bias grade of either high risk of bias, or some concerns of bias or low risk of bias. This comprehensive rating was determined through the evaluation of five domains: bias arising from the randomization process, bias due to deviations from intended interventions, bias due to missing outcome data, bias in measurement of the outcome, and bias in selection of the reported result.

For non-RCTs, the Risk of Bias In Non-randomized Research of Interventions (ROBINS-I) tool was employed [30]. This tool evaluates seven domains of bias, encompassing bias due to confounding, bias in selection of participants, bias in classification of interventions, bias due to deviations from intended interventions, bias due to missing data, bias in measurement of outcomes, and bias in the selection of reported results. Signaling questions aid in evaluating each domain, with the risk of bias for each study being classified as low, moderate, serious, or critical.

The certainty of the evidence was analyzed and summarized following the guidelines outlined in the GRADE handbook [31]. GRADE evaluates the certainty of the evidence for each specific outcome as very low, low, moderate, or high. Two reviewers (ND and DH) independently evaluated the risk of bias for each selected trial. Any differences of opinion were recorded and discussed within the research team until an agreement was reached.

### 2.5. Statistical Analyses

Consistent with the guidelines outlined in the Cochrane Handbook [32], meta-analyses can be conducted with a minimum of two studies [33]. Consequently, we conducted meta-analyses in cases where two or more studies provided baseline and follow-up data for the same measure. Mean and standard deviation data were employed to compute ESs (i.e., Hedges’ g) for tennis service outcomes in both the motor imagery and control groups. A random-effects model was used to account for the anticipated heterogeneity among studies [34,35]. The ES values were presented with 95% confidence intervals (CIs) and interpreted based on the following scale: less than 0.2 was considered trivial, 0.2 to 0.6 was small, 0.6 to 1.2 was moderate, 1.2 to 2.0 was large, 2.0 to 4.0 was very large and greater than 4.0 was extremely large [36]. In trials with multiple intervention groups, the sample size of the control group was divided proportionately to facilitate equitable comparisons among all subjects [37]. We evaluated study heterogeneity using I2 statistics. Values below 25% indicated low heterogeneity, 25–75% suggested moderate heterogeneity, and above 75% reflected high heterogeneity [38]. We utilized the extended Egger’s test to assess the studies’ publication bias risk [39]. A sensitivity analysis was performed when Egger’s test showed a significant outcome.

## 3. Results

### 3.1. Risk of Bias in Individual Studies and Certainty of the Evidence

RoB-2 assessments were conducted on four RCTs [23,40,41,42], while ROBINS-I assessments were employed for five non-RCTs [22,24,25,43,44]. Out of these trials, only three displayed a low risk of bias, while six demonstrated an overall moderate risk of bias or some concerns, as illustrated in Figure 2. The results of the RoB-2 assessments are depicted in Figure 2A. Among these RCTs, only one reported their randomization sequence generation method [23]. In contrast, the randomized procedures in the remaining three papers were not comprehensively described [40,41,42]. A visual depiction of the findings of ROBINS-I assessments is shown in Figure 2B. Three non-RCTs [22,24,43] indicated a moderate risk of bias attributed to concerns about the selection of reported outcomes. Two studies [24,43] demonstrated a moderate risk of bias due to confounding. Furthermore, one trial [43] exhibited a moderate risk of bias in selecting study participants.

The results obtained through the GRADE analysis are presented in Table 1. The GRADE assessments indicated evidence supporting the outcomes, with levels of certainty ranging from very low to low.

### 3.2. Study Characteristics

Table 2 summarizes the participants’ characteristics and the motor imagery programs utilized. The data used in the meta-analysis, extracted from the original articles, can be found in Appendix A. Publications were released between 1998 and 2023. Across all the studies included, there were a total of 548 participants. The participants’ ages ranged from 9 to 18 years old, and the sample sizes of the research groups varied from 12 to 48 individuals. Out of the nine studies, one focused on females [43], five examined males [22,23,24,40,41], and three investigated mixed gender [25,42,44]. The length of motor imagery programs was documented in nine studies, with an average of 10.9 ± 8.0 weeks (range: 4–24). The number of motor imagery sessions per week was reported in nine studies, with an average of 2.2 ± 0.8 sessions (range: 1–3). Among these, seven studies provided information on the duration of each motor imagery session, averaging 13.3 ± 6.8 min (range: 3–25), while two studies [42,44] did not report the duration of motor imagery session.

### 3.3. Effects of Motor Imagery Training on Service Speed

Seven studies were included in the service speed meta-analysis, including ten experimental groups and seven control groups (pooled *n* = 192). The motor imagery training showed no change (*p* = 0.076) in service speed with small effect (ES = 0.25, 95% CI = −0.03–0.54; Egger’s test *p* = 0.392; Figure 3) and low heterogeneity (I^2^ = 0.00%).

### 3.4. Effects of Motor Imagery Training on Service Accuracy

Eight studies were included in the service accuracy meta-analysis, including twelve experimental groups and eight control groups (pooled *n* = 237). Greater values of service accuracy (*p* < 0.001) were observed after motor imagery training (*p* = 0.007) with moderate effect (ES = 0.99; 95% CI = 0.68–1.30; Egger’s test *p* = 0.368; Figure 4) and moderate heterogeneity (I^2^ = 26.34%).

### 3.5. Effects of Motor Imagery Training on Service Return Accuracy

The service return accuracy meta-analysis comprised two studies involving three experimental groups and two control groups (pooled *n* = 78). The motor imagery training showed no change (*p* = 0.86) in service return accuracy with trivial effect (ES = 0.04; 95% CI = −0.40–0.48; Egger’s test *p* = 0.472; Figure 5) and low heterogeneity (I^2^ = 0.00%).

### 3.6. Effects of Motor Imagery Training on Service Technique

The service technique meta-analysis comprised two studies, involving three experimental groups and two control groups (pooled *n* = 41). The motor imagery training showed an increase (*p* = 0.003) of service technique with moderate effect (ES = 0.97; 95% CI = 0.33–1.61; Egger’s test *p* = 0.487; Figure 6) and low heterogeneity (I^2^ = 0.00%).

## 4. Discussion

This systematic review provides evidence that motor imagery training can improve measures of service accuracy and technique in tennis players. However, its impact on service speed and return accuracy was not found to be statistically significant. The GRADE evaluation found that the level of the evidence for the examined outcomes varied from very low to low.

The initial findings of this review indicated that motor imagery training did not affect service speed. The results could be elucidated by the observation that the enhancement of stroke speed is typically achieved through technical advancement and/or strength growth [45,46]. Motor imagery training was not intended to influence these aspects directly [25]. Nevertheless, conflicting outcomes exist in the literature, as some studies indicate that motor imagery enhances strength, mainly when movements are controlled by extensive cortical regions in the primary motor cortex [47,48]. Moreover, motor imagery is inclined to generate subtle yet potentially significant improvements in performance, especially when applied to enhance or rectify specific technical elements of motor skills [15]. Furthermore, according to some authors, the limited improvement in service speed might stem from an insufficient distribution of the weeks assigned to practicing each aspect of tennis service skills [42]. Challenging movements from a technical standpoint can potentially slow down the imaging process, extend the duration of imagery trials, and increase the risk of inaccuracies in imaging speeds, causing a retroactive interference effect that negatively impacts performance [49,50]. Robin and colleagues suggest that prolonging the acquisition phase, which consists of 20 sessions spread across three months, might be essential to significantly increase the service speed of tennis players [42]. Notably, while some investigators have noted no alteration in service speed among skilled players following motor imagery training [24,42], others, as indicated by Robin et al. [44], have demonstrated improvements in novice tennis players. This disparity in results could be explained by a potentially more limited scope for improvement among skilled players compared to beginners [42].

The current findings indicate that motor imagery training had a positive effect on service accuracy. The findings are consistent with previous research demonstrating the benefits of mental training in improving tennis shot accuracy [51]. Internal visual imagery positively impacts the acquisition and execution of tasks that rely heavily on perception for accurate performance [52]. In tasks of this nature, engaging in mental rehearsal of actions helps participants anticipate environmental changes more effectively and envision their own actions under specific conditions [53]. Moreover, neural adaptations induced by motor imagery training may improve muscle coordination, such as a decrease in activity of antagonist muscles when agonist muscles are applied [54,55], which may explain the increase in tennis service accuracy in the motor imagery training group. Interestingly, in this review, Guillot and colleagues found that imagining the tennis service with a placebo racket resulted in higher service accuracy scores compared to motor imagery training alone [25]. This may be attributed to the placebo racket enhancing visualization and promoting relaxation before serving, indicating that players’ belief in additional benefits helped alleviate anxiety during the serving process [25]. Notably, the study by Guillot et al. [25] reports that players’ perceptions of their serve quality significantly improved after motor imagery practice, and this enhancement was also observed in the placebo group. Some other researchers have indicated that elite sports performance fundamentally relies on the intricate interplay of brain functions, with the perception-action cycle being a prime example [56]. The optimal coupling between perception and action could potentially explain the improved service accuracy of tennis players, as some studies in our review have integrated motor imagery with physical practice [22,23,24,43,44]. However, these studies did not provide information on players’ sensory and perceptual experiences. Hence, the role of the perception-action cycle in such combination interventions warrants further research. Furthermore, sports practitioners and coaches commonly employ motor imagery training, either independently or as part of mental training programs integrated with other cognitive skills (e.g., concentration, self-talk, or relaxation) [6,57]. In a study conducted by Robin et al. [42], combining self-talk with motor imagery indicated that the improvement in tennis service accuracy was comparable to that achieved through motor imagery training alone. This similarity in improvement may be attributed to a plateau effect [58], which can be attributed to the participants’ expertise level (i.e., skilled tennis players) limiting further progress. Overall, the present review offers evidence supporting the positive effects of mental imagery on the service accuracy of tennis players.

Drawing from the findings of the current meta-analysis, our analysis reveals that motor imagery training did not impact a more intricate and externally oriented skill (i.e., service return accuracy). When returning a service, receivers confront tight time constraints, with services reaching high speeds and exhibiting variations in trajectory (e.g., amplitude, spin, and direction). The receiver must execute a precise shot to thwart the serving pair from strategically placing the ball in an advantageous area [59]. It seems that this mental representation would not be helpful for improved performance when the movement calls for several factors to be considered [22]. Moreover, previous investigations have shown that variances in individual visualization capacity might influence the difficulty level in learning complex motor actions [60,61]. Zapała et al. [62] found that elevated visual motor imagery scale scores were associated with quicker reaction times and improved coordination in the study group. Similarly, the findings from Robin et al.’s [24] study suggest that imagery ability plays a crucial role in determining the effectiveness of motor imagery in enhancing service return performance.

Present results show that motor imagery training significantly improved the service technique in tennis players. The positive impact of motor imagery may stem from either its motivational influence or cognitive processes that complement those generated by physical practice [50,63]. Motor imagery training for improving tennis service technique involves mental rehearsal and visualization of the critical components of the service motion [64]. During motor imagery, the brain activates neural pathways similar to those during actual physical execution [65], reinforcing the motor patterns associated with each stage of the service. Consequently, this mental practice contributes to overall technical proficiency, translating into an improved tennis service technique. Simonsmeier et al. [18] conducted a meta-analysis, revealing a positive association between motor imagery training and significant improvements in motor learning and performance. Hence, mental imagery may be an effective strategy for enhancing the service technique of tennis players. Given the restricted number of experimental trials on this aspect, additional research is required to obtain more definitive results.

Some limitations are acknowledged. Firstly, a limited number of studies (*n* = 9) were reviewed. Secondly, the impact of moderator variables (e.g., players’ age, sex) on tennis service measures could not be ascertained due to the limited number of studies. Thirdly, some trials lacked a comprehensive description of the training protocol. For example, the weekly session time for the intervention was not specified in two of the nine articles [42,44]. Fourthly, even though pooling included studies that measured the same outcome, these studies were somewhat heterogeneous with regard to training protocols (e.g., length, motor imagery combined with physical practice or implemented independently). It is possible that training-related factors might also play a role in and contribute to the effectiveness observed in motor imagery programs. Finally, the review solely considered English publications, potentially restricting the comprehensiveness of the results.

The findings of this review have practical implications for tennis practitioners, coaches, and players. The review to date has contributed to a better understanding of the relationship between motor imagery training and tennis service performance. However, limited research hinders the formulation of specific suggestions. In light of the current meta-analysis, a few suggestions for future research can be proposed. Firstly, the review only incorporates one study specifically focusing on female tennis players [43], highlighting a potential gap for future research to investigate the impact of motor imagery on this particular group. Secondly, future investigations should offer improved insight into the relationship between imagery ability and the service performance of tennis players. Thirdly, as motor imagery training programs are not homogeneous and include a variety of training modalities, further studies are needed to examine the effects of different motor imagery approaches on tennis service performance outcomes.

## 5. Conclusions

The present systematic review and meta-analysis showed the positive effects of motor imagery training in improving service accuracy and technique in tennis players. However, it appears to have no significant effect on service speed or return accuracy. Given the limited number of related studies and the evidence ranging from very low to low certainty, caution is advised in interpreting these results. Therefore, more experimental work is needed to obtain stronger conclusions.

## Figures and Tables

**Figure 1 behavsci-14-00207-f001:**
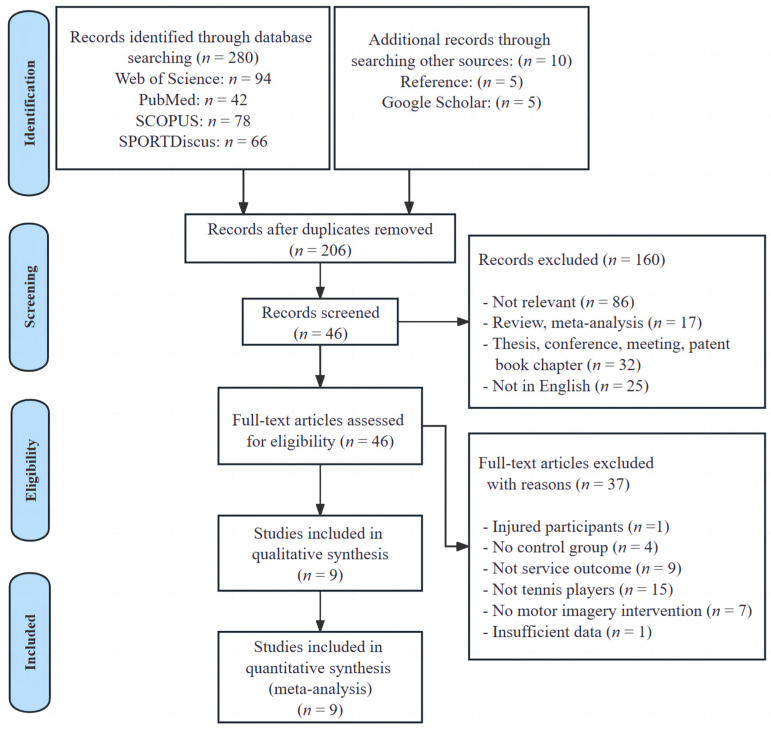
PRISMA flow diagram.

**Figure 2 behavsci-14-00207-f002:**
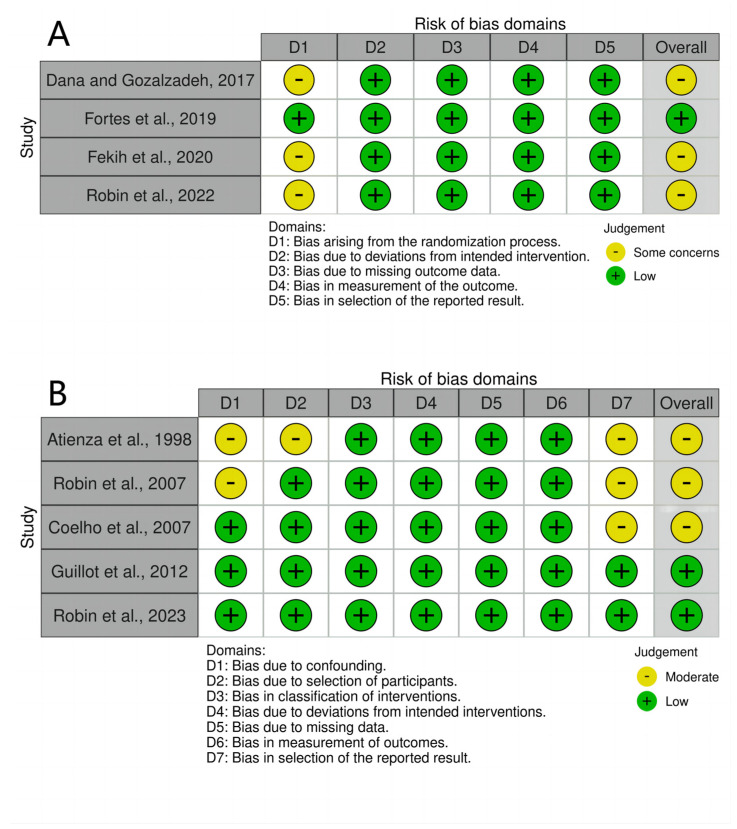
(**A**): RoB–2 assessments [23,40,41,42], (**B**): ROBINS–I assessments [22,24,25,43,44]. Created using the Robvis tool.

**Figure 3 behavsci-14-00207-f003:**
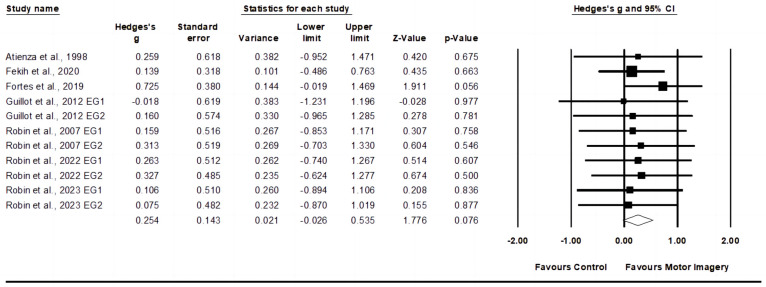
Forest plot and effect sizes for motor imagery training compared with controls for service speed [23,24,25,41,42,43,44]. EG = experimental group.

**Figure 4 behavsci-14-00207-f004:**
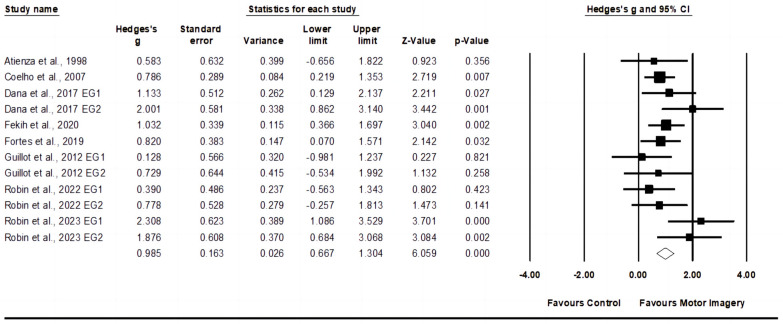
Forest plot and effect sizes for motor imagery training compared with controls for service accuracy [22,23,25,40,41,42,43,44]. EG = experimental group.

**Figure 5 behavsci-14-00207-f005:**
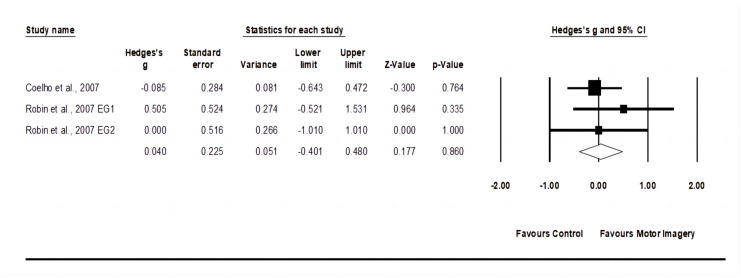
Forest plot and effect sizes for motor imagery training compared with controls for service return accuracy [22,43]. EG = experimental group.

**Figure 6 behavsci-14-00207-f006:**
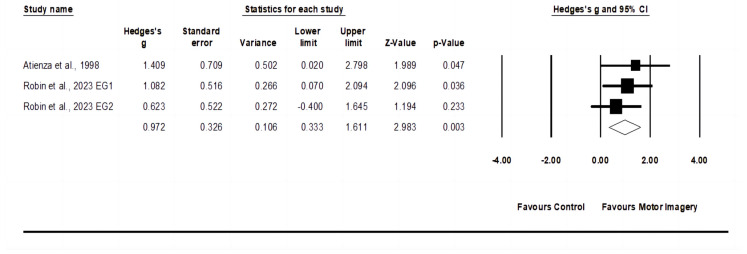
Forest plot and effect sizes for motor imagery training) compared with controls for service technique [43,44]. EG = experimental group.

**Table 1 behavsci-14-00207-t001:** GRADE analyses.

Outcomes	Certainty Assessment					No of Participants and Studies	Certainty of the Evidence (GRADE)
	Risk of Bias	Inconsistency	Indirectness	Imprecision	Risk of Publication Bias		
Service speed follow-up: range 4 to 24 weeks	Serious ^a^	Not serious	Not serious	Serious ^b^	Not serious	192 (7 studies)	⨁⨁◯◯LOW
Service accuracy follow-up: range 4 to 24 weeks	Serious ^a^	Not serious	Not serious	Serious ^b^	Not serious	237 (8 studies)	⨁⨁◯◯LOW
Service return accuracy follow-up: 8 weeks	Serious ^a^	Not serious	Not serious	Serious ^b^	Not serious	78 (2 studies)	⨁⨁◯◯LOW
Service technique follow-up: 24 weeks	Serious ^a^	Not serious	Not serious	Serious ^b^	Not serious	41 (2 studies)	⨁◯◯◯VERY LOW

GRADE, Grading of Recommendations Assessment, Development and Evaluation; ^a^ Downgraded by one level due to overall moderate or some concerns risk of bias; ^b^ Downgraded by one level, as < 400 participants were available for a comparison or a wide confidence interval (CI) around the effect estimate; we considered a CI to be wide if it included both a small (0.2–0.6) and large effect size (>1.2–2.0). Downgraded by two levels in case of imprecision based on both assessed points. GRADE Working Group grades of evidence: High certainty: we are very confident that the true effect lies close to that of the estimate of the effect. Moderate certainty: we are moderately confident in the effect estimate: the true effect is likely to be close to the estimate of the effect, but there is a possibility that it is substantially different. Low certainty: our confidence in the effect estimate is limited: the true effect may be substantially different from the estimate of the effect. Very low certainty: we have very little confidence in the effect estimate, the true effect is likely to be substantially different from the estimate of effect.

**Table 2 behavsci-14-00207-t002:** Characteristics of the studies examined in the present review.

Study		Population Characteristics			Intervention	Type of Exercise	Outcome (s)
	*n*	Sex	Age	Level			
Atienza et al., 1998 [43]	12	Female	9–12 years	Tennis school	Freq: 2 times/week Time: 15 min Length: 24 weeks	EG1: physical practice +video EG2: physical practice +video + imagery training CG: physical practice	EG1: service speed ↑, service accuracy →, Service technique ↑; EG2: Service speed →, service accuracy →, service technique ↑
Robin et al., 2007 [24]	30	Male	19 ± 2.5 years	Regional or national level ≥7 years	Freq:1 time/week Time: 3 min Length: 8 weeks	EG1: motor imagery + physical training (good imagers) EG2: motor imagery + physical training (poor imagers) CG: reading a magazine	EG1: service speed →, service return accuracy ↑; EG2: service speed →, service return accuracy ↑
Coelho et al., 2007 [22]	48	Male	16–18 years	National	Freq: 3 times/week Time: 25 min Length: 8 weeks	EG: imagery + technical practice CG: technical practice	Service accuracy ↑, service return accuracy →
Guillot et al., 2012 [25]	22	Mixed	EG1: 14.25 ± 2.60 years EG2:14.43 ± 3.05 years CG: 16.29 ± 5.50 years	3.0 ± 1.2 years	Freq: 2 times/week Time: 15 min Length: 6 weeks	EG1: motor imagery EG2: imagery + placebo racket CG: physical practice	EG1: service speed →, service accuracy ↑; EG2: service speed →, service accuracy ↑
Dana & Gozalzadeh, 2017 [40]	36	Male	15–18 years	Novices	Freq: 3 times/week Time: 15 min Length: 6 weeks	EG1: internal imagery EG2: external imagery CG: physical practice	EG1: service accuracy ↑; EG2: service accuracy ↑
Fortes et al., 2019 [23]	28	Male	15–16 years	≥2 years	Freq: 3 times/week Time: 10 min Length: 8 weeks	EG: motor imagery + physical/technical training CG: videos (Olympic history) + physical/technical training	Service speed ↑, service accuracy ↑
Fekih et al., 2020 [41]	38	Male	EG: 16.9 ± 0.6 years CG: 16.7 ± 0.8 years	Tennis clubs EG: 5.4 ± 1.3 years CG: 5.7 ± 1.2 years	Freq: 3 times/week Time: 10 min Length: 4 weeks	EG: motor imagery CG: videos about the history of the Olympic Games	Service speed →, service accuracy↑
Robin et al., 2022 [42]	33	Mixed	15.9 ± 2.1 years	Regional and national 9.5 ± 1.8 years	Freq: 2 times/week Time: NR Length: 10 weeks	EG1: motor imagery EG2: motor imagery+ self-talk CG: physical practice	EG1: service speed →, service accuracy ↑; EG2: service speed →, service accuracy ↑
Robin et al., 2023 [44]	33	Mixed	9–13 years	Novice 1–2 years	Freq: 1 time/week Time: NR Length: 24 weeks	EG1: motor imagery + physical training (good imagers) EG2: motor imagery + physical training (poor imagers) CG: reading a magazine	EG1: service speed ↑, service accuracy ↑, service technique ↑; EG1: service speed ↑, service accuracy ↑, service technique ↑

EG, experimental group; CG, control group; years, years; Freq, frequency; NR, not reported.

## Data Availability

The datasets generated and analyzed for this study can be requested by correspondence authors.

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
