# Peer review of "Does Motor Imagery Training Improve Service Performance in Tennis Players? A Systematic Review and Meta-Analysis"

_behavsci, 2024, doi:10.3390/bs14030207_

Round 1

Reviewer 1 Report

Comments and Suggestions for Authors

The present paper is very interesting and is well-structured. Moreover, it is seemed to be pertinent to highlight the premise of this manuscript is a worthy one, and the authors spent a great time in the research and writing. However, there are a few issues that need to be addressed to the manuscript prior to publication. Please note that these corrections/suggestions should not be seen as a negative against the hard work the authors have put into this manuscript.

Abstract

It is relevant if there is a background included in the abstract. 

It is suggested to emphasize the pertinency of the study. 

The authors should highlight the importance of the results obtained through their work. 

Introduction 

The section is well-written and clear, conducting in a simple way to the relevance of the study purpose. However, it is suggested to include the specific sport that is analyzed in this systematic review when the authors exposed the study's purpose.

Methodology section is appropriated and well-developed.

Results are interesting and the authors presented them in a very enlightening way.

Discussion is well exposed and highlight the main results obtained with an argumentative power. 

Conclusion 

In this section the authors should include to whom professionals is the results of the systematic review important. The results obtained will be a useful information for who? The authors should reinforce it. 

Author Response

Comments

Abstract

1. It is relevant if there is a background included in the abstract.

 Answer: we have added a short background in the abstract. Line 10-12.

2. It is suggested to emphasize the pertinency of the study.

Answer: thank you for your suggestion, we have revised the abstract to emphasize that while motor imagery training is widely adopted by tennis players and coaches for learning and performance enhancement, its impact on service performance in tennis players remains uncertain. Lines 10-12.

3. The authors should highlight the importance of the results obtained through their work.

Answer: thank you for your comments. Taking into account your feedback and the suggestions of other reviewers, we have updated the abstract section. While motor imagery training has demonstrated a positive effect on service accuracy and technique in tennis players, the limited number of related studies and the very low to low certainty of the available evidence suggest that these results should be approached with caution. We advocate for further research on this topic to draw more definitive conclusions.

Introduction

1. The section is well-written and clear, conducting in a simple way to the relevance of the study purpose. However, it is suggested to include the specific sport that is analyzed in this systematic review when the authors exposed the study's purpose.

Answer: thank you for your review. We revised the statement of the purpose of the study. Line 67-68.

Conclusion

1. In this section the authors should include to whom professionals is the results of the systematic review important. The results obtained will be a useful information for who? The authors should reinforce it.

Answer: thank you for your valuable suggestion; we have underscored the practical significance of this review in the practice application part. Lines 347-350.

Reviewer 2 Report

Comments and Suggestions for Authors

Many thanks to the editors for considering me for the review of this work and thanks to the authors for this interesting work. A number of comments are made below in order to contribute to the paper.

Regarding the abstract the authors should remove the data to make it easier to read. More detailed data will be found in the results section.
As for the introduction, it is considered to be very good reading and guides the reader to the important variables in the study. It is recommended (it is not an obligation) that the authors also mention the importance of imagery training in the stages of sports training in a more profound way. The introduction is very short, so the authors could go into more depth in other ways also related to performance.

In the method the authors indicate that they use the PRISMA methodology. This should be indicated in the abstract as well. It is recommended that section 3.1 of the results as well as the flow diagram be part of the methodology.
The discussion shows the lack of balance between the introduction and this section. It can be seen that a good analysis of the state of the art has been made but it highlights the need to improve the introduction. On the other hand, it is recommended that points 5 and 6 form part of the discussion.

Author Response

Comments

1. Regarding the abstract the authors should remove the data to make it easier to read. More detailed data will be found in the results section.

Answer: thank you for your suggestion. We have removed the data in the abstract. Lines 19-20.

2. As for the introduction, it is considered to be very good reading and guides the reader to the important variables in the study. It is recommended (it is not an obligation) that the authors also mention the importance of imagery training in the stages of sports training in a more profound way. The introduction is very short, so the authors could go into more depth in other ways also related to performance.

Answer: thank you for your valuable comments. We have incorporated the theory of motor imagery into our introduction to help readers better grasp its role in enhancing performance. This, in turn, underscores the importance of integrating this type of training into sports training programs. Lines 36-44.

3. In the method the authors indicate that they use the PRISMA methodology. This should be indicated in the abstract as well. It is recommended that section 3.1 of the results as well as the flow diagram be part of the methodology.

Answer: thank you for your careful review; we have revised the abstract and added section 3.1 to section 2.1. Lines 13-14, line 90.

4. The discussion shows the lack of balance between the introduction and this section. It can be seen that a good analysis of the state of the art has been made but it highlights the need to improve the introduction. On the other hand, it is recommended that points 5 and 6 form part of the discussion.

Answer: thank you for your valuable comments. According to your feedback and other reviewers' comments, We have revised the introduction section, lines 36-58. Additionally, we have included points 5 and 6 in the discussion section. Lines 335-358.

Reviewer 3 Report

Comments and Suggestions for Authors

lines 37-8-  if you do not identify what psychological and physiological factors, or if they are not important for the focus of the  systematic review and meta-analysis, consider to remove

lines 39-40- it is not consensual that "(...) motor imagery and the execution of real motor tasks involve similar neuronal components at both cortical and subcortical levels (...)". present studies that contradict this statement.

lines 47-8- visualizing previous actions is not in the scope of motor imagery, consider to remove

lines 52-3- dynamic motor imagery is a different approach than the one defined in line 35. clearly explain the differences 

lines 65-66- service is a discrete, precise, fast movement, without direct opponent pressure; meaning that, probably, this meta-analysis will only contribute contribute for motor actions with similar constraints in other sports, probably...

lines 85-87- explain the sustainability of the following criteria:

- Motor imagery training interventions were required to be a minimum of two weeks in duration

- the study involved one or more tennis service performance outcomes

Also in "Eligibility Criteria"- and what about motor imagery protocols? (see Table 2). with a such diversity of protocols, how is it possible to measure the contribution of motor imagery alone or combined? 

Grade Analysis (Table 1) and Conclusions- if risk of bias and imprecision are serious, resulting in grade low or very low, how can you state that "(...) motor imagery training has a substantial positive effect on improving service accuracy and service technique (...)"?

lines 241-2- if you have a great diversity of protocols (some with only imagery, others mixed with practice), how can you state that  mental training improves tennis shot accuracy? and what about the contribution of perception-action cycles during practice? and what about sensory and perceptual information, including with placebo (ref. 23)?

Author Response

Comments

1. lines 37-8-  if you do not identify what psychological and physiological factors, or if they are not important for the focus of the systematic review and meta-analysis, consider to remove.

Answer: thank you for your suggestion, we have removed this sentence. please see the introduction section.

2. lines 39-40- it is not consensual that "(...) motor imagery and the execution of real motor tasks involve similar neuronal components at both cortical and subcortical levels (...)". present studies that contradict this statement.

Answer: thank you for your comment. We have removed the sentence and revised this paragraph, incorporating additional information to justify the importance of motor imagery for enhancing performance. Lines 36-44.

3. lines 47-8- visualizing previous actions is not in the scope of motor imagery, consider to remove.

Answer: thank you for your careful reading. We have removed this sentence; please see the introduction section.

4. lines 52-3- dynamic motor imagery is a different approach than the one defined in line 35. clearly explain the differences

Answer: thank you for pointing this out; we have removed this reference and changed it to another example in order to be consistent with the definition of motor imagery, lines 55-57.

5. lines 65-66- service is a discrete, precise, fast movement, without direct opponent pressure; meaning that, probably, this meta-analysis will only contribute for motor actions with similar constraints in other sports, probably...

Answer: thank you for your comments, we have revised our statement to specifically focus on tennis service outcomes. Lines 71-72.

6. lines 85-87- explain the sustainability of the following criteria:

6.1. Motor imagery training interventions were required to be a minimum of two weeks in duration

Answer: thank you for your valuable comments. We have removed this statement and have carefully reviewed the full texts as shown in Figure 1. We found no study of less than 2 weeks in duration; all studies are 4 weeks or longer (see Table 1), hence, we decided to delete it.

6.2. The study involved one or more tennis service performance outcomes

Answer: thank you for your review. We have rewritten this statement to make it clearer to the reader. Line 108. Our focus is on investigating the impact of motor imagery on service performance; therefore, the studies we select must examine the outcomes of tennis services.

7. Also in "Eligibility Criteria"- and what about motor imagery protocols? (see Table 2). with a such diversity of protocols, how is it possible to measure the contribution of motor imagery alone or combined?

Answer: thank you for pointing this out. We have revised the Eligibility Criteria section to outline the inclusion criteria for the training protocol. Lines 103-106. Additionally, due to the scarcity of studies, evaluating the potential impact of training factors on the effects of motor imagery training is not feasible. Therefore, we have recognized the diversity of protocols as a limitation in this review and emphasized that further studies are necessary to investigate the effects of different motor imagery protocols on tennis service performance outcomes. Lines 341-343, lines 356-358.

8. Grade Analysis (Table 1) and Conclusions- if risk of bias and imprecision are serious, resulting in grade low or very low, how can you state that "(...) motor imagery training has a substantial positive effect on improving service accuracy and service technique (...)"?

Answer: thank you for your comments. We have revised our conclusion to highlight that the limited number of studies, along with the very low to low certainty of evidence from the included studies, means that the results need to be interpreted with caution. Lines 360-365.

9. lines 241-2- if you have a great diversity of protocols (some with only imagery, others mixed with practice), how can you state that mental training improves tennis shot accuracy? and what about the contribution of perception-action cycles during practice? and what about sensory and perceptual information, including with placebo (ref. 23)?

Answer: thank you for your review. We have revised this section to include a discussion on the potential benefits of perception-action cycles during practice. Additionally, we have added information about sensory and perceptual information in reference 23 (i.e., Guillot et al., 2012). However, this study only mentioned players’ perceptions of their service quality. Lines 297-307. In addition, we have recognized the diversity of protocols as a limitation in this review and added a statement advising that the results need to be interpreted with caution in the conclusion section. Lines 360-365, lines 341-343.

Reviewer 4 Report

Comments and Suggestions for Authors

The introduction is well-structured, with clear delineation of the importance of the service shot, the challenges associated with mastering it, and the potential role of motor imagery training in addressing these challenges.

While the paper mentions specific databases and search terms used, providing a more detailed description of the search strategy, including any filters or limits applied, would improve transparency and reproducibility. Additionally, specifying the date range of the search and any language restrictions would add clarity.

While the paper mentions the use of established tools (RoB-2 for RCTs and ROBINS-I for non-RCTs) for assessing risk of bias, providing more specific details on how each domain of bias was evaluated and any criteria used for judgment would improve transparency and allow readers to better understand the validity of the included studies.

The Discussion section effectively synthesizes the findings of the systematic review on motor imagery training in enhancing service performance among tennis players. It provides a balanced interpretation of the results, highlighting both the positive effects observed and the limitations of the study.

I appreciate the thorough examination of the potential mechanisms underlying the observed improvements in service accuracy and technique, including the role of mental rehearsal, neural adaptations, and the complementarity of mental and physical practice. The discussion of factors influencing the effectiveness of motor imagery training, such as individual differences in visualization capacity and expertise level, adds depth to the analysis.

Furthermore, the acknowledgment of study limitations, such as the small number of studies reviewed and the lack of detailed training protocols in some cases, demonstrates transparency and helps contextualize the findings. The discussion of practical implications for coaches, including recommendations for incorporating motor imagery training into training programs, adds value for practitioners in the field. Overall, the Discussion section effectively contextualizes the research findings, explores their implications, and identifies avenues for future research. It provides a solid conclusion to the systematic review, contributing to the broader understanding of motor imagery training in sports performance.

Author Response

Suggestions

1. While the paper mentions specific databases and search terms used, providing a more detailed description of the search strategy, including any filters or limits applied, would improve transparency and reproducibility. Additionally, specifying the date range of the search and any language restrictions would add clarity.

Answer: Thank you for your careful review. Table S1 contains the search strings (including filters and dates) used for each database. Please see the attachment. Moreover, we have stated that journal articles in English, from the databases' inception until December 6, 2023, were searched. Line 84

2. While the paper mentions the use of established tools (RoB-2 for RCTs and ROBINS-I for non-RCTs) for assessing risk of bias, providing more specific details on how each domain of bias was evaluated and any criteria used for judgment would improve transparency and allow readers to better understand the validity of the included studies.

Answer: thank you for your comments. We have added a detailed description of each domain of bias. Lines127-140.

Round 2

Reviewer 3 Report

Comments and Suggestions for Authors

No further comments